# Large socioeconomic gap in period life expectancy and life years spent with complications of diabetes in the Scottish population with type 1 diabetes, 2013–2018

Andreas Höhn[1,2]*, Stuart J. McGurnaghan[2], Thomas M. Caparrotta[2], Anita Jeyam[2], Joseph E. O'Reilly[2], Luke A. K. Blackbourn[2], Sara Hatam[2], Christian Dudel[3], Rosie J. Seaman[4], Joseph Mellor[5], Naveed Sattar[6], Rory J. McCrimmon[7], Brian Kennon[8], John R. Petrie[6], Sarah Wild[5], Paul M. McKeigue[5], Helen M. Colhoun[2,9], on behalf of the SDRN-Epi Group[¶]

1 School of Geography and Sustainable Development, The University of St. Andrews, St. Andrews, United Kingdom, 2 Institute of Genetics and Cancer, The University of Edinburgh, Edinburgh, United Kingdom, 3 Max Planck Institute for Demographic Research, Laboratory of Population Health, Rostock, GER, 4 MRC/CSO Social and Public Health Sciences Unit, University of Glasgow, Glasgow, United Kingdom, 5 Usher Institute of Population Health Sciences and Informatics, The University of Edinburgh, Edinburgh, United Kingdom, 6 Institute of Cardiovascular and Medical Sciences, University of Glasgow, Glasgow, United Kingdom, 7 School of Medicine, University of Dundee, Dundee, United Kingdom, 8 Queen Elizabeth University Hospital, University Glasgow, Glasgow, United Kingdom, 9 Public Health, NHS Fife, Kirkcaldy, United Kingdom

¶ Membership of the Scottish Diabetes Research Network Epidemiology Group is listed in the Acknowledgments.
* ah378@st-andrews.ac.uk

**Data Availability Statement:** The analysed data were provided de-identified, with approval from the Public Benefit and Privacy Panel (PBPP refs. 1617-

## Abstract

### Background

We report the first study to estimate the socioeconomic gap in period life expectancy (LE) and life years spent with and without complications in a national cohort of individuals with type 1 diabetes.

### Methods

This retrospective cohort study used linked healthcare records from SCI-Diabetes, the population-based diabetes register of Scotland. We studied all individuals aged 50 and older with a diagnosis of type 1 diabetes who were alive and residing in Scotland on 1 January 2013 (N = 8591). We used the Scottish Index of Multiple Deprivation (SIMD) 2016 as an area-based measure of socioeconomic deprivation. For each individual, we constructed a history of transitions by capturing whether individuals developed retinopathy/maculopathy, cardiovascular disease, chronic kidney disease, and diabetic foot, or died throughout the study period, which lasted until 31 December 2018. Using parametric multistate survival models, we estimated total and state-specific LE at an attained age of 50.

0147), originally set up under Privacy Advisory Committee (PAC) 33/11, with approval from the Scotland A Research Ethics Committee (ref. 11/AL/0225). NHS data governance rules do not permit us to secondarily share the analysed data directly. However, bona fide researchers can apply to the Scottish Public Benefits Protection Committee for access to these data or get in contact with the Scottish Diabetes Research Network in order to gain further information on the data and legislations surrounding the access to the data. The Scottish Diabetes Research Network can be contacted via the following address: Diabetes Support Unit, Level 8, Ninewells Hospital, Dundee DD1 9SY, email: administrator-sdrn@dundee.ac.uk. The Scottish Public Benefits Protection Committee provided detailed information and instructions on data access requests here: https://www.informationgovernance.scot.nhs.uk/pbpphsc/ and be contacted via the following email address: phs.edris@phs.scot. For further information on NHS Scotland diabetes data provisioning see: https://www.nhsresearchscotland.org.uk/research-areas/diabetes/training-and-events. The corresponding author can be contacted for any further genuine requests to audit the validity of the analyses. We are happy to share summary statistics for those wishing to conduct meta-analyses with other studies.

**Funding:** This study was supported by funding from Diabetes UK. In particular the following grants: 17/0005627 - received by HMC 8/0005786 - received by TMC. The funder had no role in designing the study or in analysing and interpreting data and results.

**Competing interests:** All authors have completed and submitted the ICMJE Form for Disclosure of Potential Conflicts of Interest. TMC reports grants from Diabetes UK Grant: 18/0005786, during the conduct of the study. SHW reports meeting attendance supported by Novo Nordisk, outside the submitted work. RJM reports personal fees from Sanofi, personal fees from NovoNordisk, outside the submitted work. NS reports personal fees from Amgen, personal fees from AstraZeneca, grants and personal fees from Boehringer Ingelheim, personal fees from Eli Lilly, personal fees from Novo Nordisk, personal fees from Pfizer, personal fees from Sanofi, outside the submitted work; HMC reports grants, personal fees and other from Eli Lilly and Company, during the conduct of the study; grants from AstraZeneca LP, other from Novartis Pharmaceuticals, grants from Regeneron, grants from Pfizer Inc, other from Roche Pharmaceuticals, other from Sanofi Aventis, grants and personal fees from Novo Nordisk, outside the

## Results

At age 50, remaining LE was 22.2 years (95% confidence interval (95% CI): 21.6 − 22.8) for males and 25.1 years (95% CI: 24.4 − 25.9) for females. Remaining LE at age 50 was around 8 years lower among the most deprived SIMD quintile when compared with the least deprived SIMD quintile: 18.7 years (95% CI: 17.5 − 19.9) vs. 26.3 years (95% CI: 24.5 − 28.1) among males, and 21.2 years (95% CI: 19.7 − 22.7) vs. 29.3 years (95% CI: 27.5 − 31.1) among females. The gap in life years spent without complications was around 5 years between the most and the least deprived SIMD quintile: 4.9 years (95% CI: 3.6 − 6.1) vs. 9.3 years (95% CI: 7.5 − 11.1) among males, and 5.3 years (95% CI: 3.7 − 6.9) vs. 10.3 years (95% CI: 8.3 − 12.3) among females. SIMD differences in transition rates decreased marginally when controlling for time-updated information on risk factors such as HbA1c, blood pressure, BMI, or smoking.

## Conclusions

In addition to societal interventions, tailored support to reduce the impact of diabetes is needed for individuals from low socioeconomic backgrounds, including access to innovations in management of diabetes and the prevention of complications.

## Introduction

Socioeconomic inequality is one of the most important factors shaping health and mortality outcomes among general populations [1] and among populations with type 1 diabetes [2]. Addressing socioeconomic inequalities in health and mortality among individuals with type 1 diabetes has been identified as one of the key priorities for diabetes care in a number of Western countries [3, 4], including Scotland [5].

In Scotland, a large socioeconomic gap has been reported for all-cause mortality [6], and with respect to all-cause mortality before age 50 [7] for the population with type 1 diabetes. Socioeconomic differences have also been identified regarding the management of type 1 diabetes, such as glucose control [8, 9] and complications, including the prevalence of diabetic neuropathy [10] and diabetic retinopathy [11]. The Scottish Government's most recent Diabetes Improvement Plan (2021–2026) stated that addressing the challenges associated with socioeconomic inequalities in access to care and health outcomes among the population with type 1 diabetes needs to be a priority [5]. To date, the magnitude of socioeconomic differences in period life expectancy (LE) and health adjusted LE for the population with type 1 diabetes in Scotland remains unknown.

LE is a cross-sectional summary measure of age-specific mortality rates. It is based on the assumption that a hypothetical cohort will be exposed to the age-specific mortality rates that were observed at one particular point in time within the population (LE) [12]. Due to its cross-sectional character and implicit age-standardization, LE estimates are widely available and commonly used to monitor socioeconomic inequalities in mortality and health [13].

Estimates of LE are available for populations with type 1 diabetes in several countries [14–16]. Using data from the Australian diabetes register, Huo et al. (2016) showed that LE at age 20 was 47.6 years for males and 51.5 years for females in the period 1997–2010 [15]. In a previous study for the Scottish population with type 1 diabetes, LE at age 20 was 46.2 years for males and 48.1 years for females in the period 2008–2010 [16]. Based on data from the Swedish

submitted work. This does not alter our adherence to PLOS ONE policies on sharing data and materials. No other disclosures were reported.

**Abbreviations:** AIC, Akaike information criterion; CHI-Number, Community Health Index—Number; CKD, chronic kidney disease; CKD-EPI, Chronic Kidney Disease Epidemiology Collaboration; CVD, cardiovascular disease; DKA, diabetic ketoacidosis; eGFR, estimated glomerular filtration rate; HR, Hazard Ratio; ISD, Information Services Division; KDIQO, Kidney Disease: Improving Global Outcomes; LE, life expectancy; NHS, National Health Service; NRS, National Records of Scotland; PY, person-years; SCI-Diabetes Database, Scottish Care Information-Diabetes Database; SIMD, Scottish Index of Multiple Deprivation.

National Diabetes Register, Petrie et al. (2016) estimated changes in LE. For people with type 1 diabetes in Sweden, LE at age 20 increased between the period 2002–2006 and the period 2007–2011 from 47.7 to 49.8 years for males and from 51.7 to 51.9 years for females [14]. All of these studies provided important insights into the levels of LE among the population with type 1 diabetes. These studies also estimated the loss in LE due to type 1 diabetes by making comparisons with LE estimates for the corresponding general populations. None of these studies quantified differences in life expectancy between socioeconomic groups for the population with type 1 diabetes.

When combined with health information, LE can be separated into years spent with and without certain complications to measure health-adjusted LE. Health-adjusted LE can provide a better reflection of the excess burden of ill health experienced by particular populations than studying only LE [17]. Very little is known about socioeconomic inequalities in health-adjusted LE for populations with type 1 diabetes. Previous real-world evidence comes from a small number of studies which focused on individuals with type 1 and type 2 diabetes in combination [18, 19]. In addition, a number of simulation studies identified particular risk-factors for disease progression and mortality. Using clinical trial data, these studies have provided evidence on the potential impact changes to treatment regimen may have on health-adjusted life years for individuals with type 1 diabetes [20–22]. To date no study has examined socioeconomic inequalities in health-adjusted LE for a real-world population with type 1 diabetes.

The goal of this retrospective cohort study was to estimate LE for males and females of the Scottish population with type 1 diabetes at age 50 for different socioeconomic groups and to estimate how many subsequent years of life were spent with and without the most common complications of diabetes. We calculated LE and health-adjusted LE for the period 2013–2018 to provide the most recent estimates. We expected that the general patterns in LE and health-adjusted LE among the population with type 1 diabetes would mirror the most common patterns observed among the general population. We hypothesized that LE would be higher among females than males with type 1 diabetes. Mirroring patterns in the general Scottish population, we expected to observe a large socioeconomic gap in LE and health-adjusted LE among the population with type 1 diabetes. We anticipated that the most socioeconomically deprived group would have a double disadvantage: the lowest LE and lowest health-adjusted LE. This would mean that, on average, individuals with type 1 diabetes from the most deprived group live the shortest lives and spend the most years living with complications of diabetes.

## Materials and methods

### Data

We used routinely collected, electronic healthcare records extracted from the Scottish Care Information-Diabetes (SCI-Diabetes) database. This database is a nationwide diabetes register for Scotland and covers >> 99% of all individuals who were assigned a diagnosis of type 1 or type 2 diabetes in primary or secondary care in Scotland [9]. Since 2004, the database has had nationwide coverage. The Information Services Division (ISD) of the National Health Service (NHS) in Scotland used the Community Health Index (CHI) Number, a unique personal identification number, to link data from the SCI-Diabetes database with information on hospital admissions and date and cause of death, provided by National Records of Scotland (NRS).

We utilized an algorithm to identify individuals with type 1 diabetes in the SCI-Diabetes database. This algorithm is based on age, drug prescriptions, and clinical information on the type of diabetes [9]. The algorithm has additional features to ensure that prescription histories don't contradict the clinical information (for example to rule out a lengthy period of diabetes before insulin and no co-prescribing of non-metformin oral diabetes drugs [23].

Data acquisition, data linkage, and the type-of-diabetes algorithm have been described in more detail in previous papers [16, 23, 24].

## Ethics approval

Data and data linkage were set up with approval from the Scottish A Research Ethics Committee (ref 11/AL/0225), Caldicott Guardians and the Privacy Advisory Committee (PAC—ref 33/11), now running with approval from the Public Benefit and Privacy Panel for Health and Social Care (PBPP—reference 1617–0147).

## Study population

We defined the study population as all individuals aged 50 and older with a diagnosis of type 1 diabetes, who were alive and residing in Scotland on 01 January 2013, and who were observable for more than 30 consecutive days (N = 8627). For this study population, sociodemographic data and information on health status were obtained and continuously updated using a long-format survival table capturing 30-day slices of time. The longitudinal healthcare records were sliced every 30 days for each individual [25]. Individuals contributed data to the follow-up starting on 01 January 2013 and were censored in case they became unobservable during the follow-up period or in case they were alive on 31 December 2018.

To determine an accurate health status for individuals at study entry, we utilized all available information prior to study entry. For this purpose, we were able to use existing data with nationwide coverage reaching back to 2004. For some individuals, data and retrospective information were available for years prior to 2004. An overview of these periods is provided in Fig 1—**Panel (A).**

## Definition of covariates and complications

For each individual, we obtained information on sex, age, and socioeconomic deprivation. To capture socioeconomic deprivation, we utilized the Scottish Index of Multiple Deprivation 2016 (SIMD 2016) [26]. The SIMD is an area-level index created by the Scottish Government. The SIMD captures area-level deprivation across multiple aspects, including unemployment,

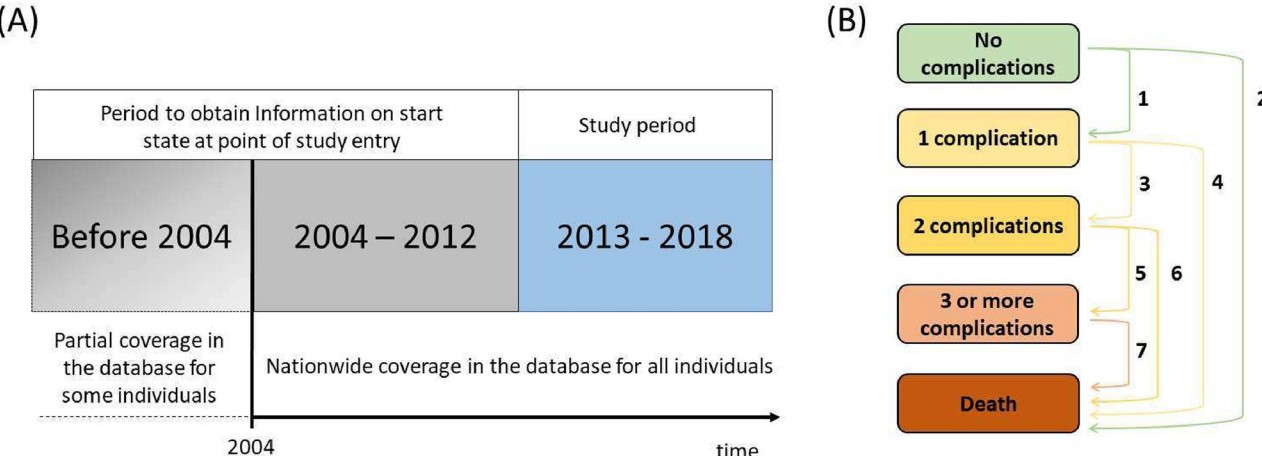

**Fig 1. Overview of the study design.** Panel (A) illustrates the captured study period and the preceding period to obtain an exact health status at study entry on 01 January 2013. Panel (B) shows how the five mutually exclusive health states—four transient states and one absorbing state—were connected via seven distinct transitions.

income, education, and crime rates of the data zone where an individual's place of usual residence is [26]. The grouping into quintiles was carried out by the Scottish Government and is based on a ranking of all 6,976 territorial data zones in Scotland, with each data zone reflecting a population size of approximately 700 people. In line with recommendations by ISD for the correct use of the SIMD, we used the SIMD 2016 release consistently throughout the study period. As we used one SIMD release consistently throughout the study period, there were no changes in SIMD quintile for the studied individuals.

We obtained information on four diabetes-related complications: retinopathy/maculopathy, cardiovascular disease (CVD), chronic kidney disease (CKD), and diabetic foot. For each condition, we used a binary code to indicate whether individuals developed a condition within any 30-day time slice. Individuals could accumulate complications but once a complication was diagnosed, we assumed it to be irreversible. For each individual, we counted the total number of complications within each 30-day slice of time. We also captured all-cause mortality.

A detailed overview of the definitions of each complication is presented in S1 Table. Information on retinopathy/maculopathy were obtained from routine eye screening data and classified based on the worst score of either the left or the right eye. We classified people based on whether they had sufficient retinal changes to warrant eye clinic referral.

Incident CVD was defined as a hospital discharge with ICD-9/10 codes for ischemic heart disease, cerebrovascular disease, hypertension, heart failure, cardiac arrhythmia, myocardial infarction, transient ischemic attack, or peripheral arterial disease as a primary or secondary diagnosis.

The presence of diabetic foot was defined based on the results of diabetic foot risk screenings. We classified individuals as having developed a diabetic foot if either foot was denoted as having a high risk or if active ulcers or amputations were present.

The Kidney Disease: Improving Global Outcomes (KDIQO) Clinical Practice Guideline for Diabetes Management in Chronic Kidney Disease defines CKD as abnormalities of kidney structure or function, present for >3 months, with implications for health [27]. Given the importance of albuminuria in predicting a more rapid renal function decline at any eGFR level CKD is further classified based on Cause, GFR category (G1–G5), and Albuminuria category (A1–A3). Here for simplicity, we define CKD as ever had a record of an estimated glomerular filtration rate (eGFR) of < 60 mL/min/1.73 m^2 or being in receipt of renal replacement therapy (i.e., KDIQO G3a or worse). We did not require the presence of albuminuria, but we provide data on the coexistence of albuminuria in the results section. Transient acute falls in eGFR were not included as CKD. All values for eGFR are based on the equation provided by the Chronic Kidney Disease Epidemiology Collaboration (CKD-EPI) using serum creatinine [28].

Missing information on individual's characteristics and health measures at study entry were imputed using multiple imputation provided in the R-package *Amelia II* [29]. To increase the accuracy of imputation, we used longitudinal information on HbA1c, total cholesterol, high-density lipoprotein-cholesterol, low-density lipoprotein-cholesterol, systolic blood-pressure, diastolic blood pressure, BMI, smoking status, as well as information on previous hospital admissions for diabetic ketoacidosis (DKA) and hypoglycemia. An overview of the share of missing information at time of study entry is presented in S2 Table.

## Multistate approach

The multistate approach in this study was conceptualized to capture the number of years lived in disease states for broad population subgroups. Similar designs have previously been used in

studies of multimorbidity for general populations [30]. For our study population, we constructed a history of transitions between five mutually exclusive states by capturing whether individuals developed new complications or died. An overview of the state space and all seven distinct transitions is provided in Fig 1 - **Panel (B)**.

The state space included the four transient states 'no complications' (state 1), '1 complication' (state 2), '2 complications' (state 3), '3 or more complications' (state 4), and one absorbing state 'death' (state 5). Depending on the number of complications at time of study entry, individuals started either in the state of 'no complication' (state 1), '1 complication' (state 2), '2 complications' (state 3), or '3+ complications' (state 4). In addition to the defined state space, Fig 1 - **Panel (B)** illustrates all allowed transitions. Individuals could remain in the current state, transition either one state up, or transition into the absorbing state 'death' throughout the follow-up period.

Transitions back to previous states were not allowed. We assumed that the onset of complications and the process of accumulating complications was an irreversible process. In addition, we did not allow transitions that directly skipped one or more transient states within one 30-day slice of time. This means that we did not allow the following transitions: 'no complication' to '2 complications' (from state 1 to state 3), 'no complications' to '3+ complications' (from state 1 to state 4), and from '1 complication' to '3+ complications' (state 2 to state 4). Out of all 8627 individuals, we only observed a total number of 36 individuals who ever transitioned two transient states, while no individuals transitioned three transient states within one 30-day time slice. As these transitions were very rare events, we excluded the affected individuals. This determined the final size of the study population of N = 8591 individuals.

## Estimation of transition-specific models

We used a continuous time Markovian multistate approach and modelled seven transitions between five states [31]. For model estimation, we modified the data as follows. We captured the time of transitions using age as a time scale and setting age 50 as time point 0. All models were estimated separately for each transition to ensure the best possible fit.

For the main results, we estimated two sets of seven parametric models. Two distinct sets of models were required to obtain estimates for two different sets of strata: (1) sex, and (2) sex and SIMD quintile. Within each set, all models have an identical functional form for every transition. An overview of the functional form of all models used for the main analysis is given in S3 Table.

The first set of models included only the covariate 'sex' to derive LE and health-adjusted LE estimates for all males and all females, not considering their SIMD quintile. The second set included the covariates 'sex' and 'SIMD quintile' to derive estimates for males and females from the five SIMD quintiles. For all models, we utilized a Gompertz distribution as this distribution consistently minimized the respective Akaike information criterion (AIC).

For the second set of models, we examined the impact of an interaction effect between sex and SIMD quintile. The interaction effect did not improve the fit of any of the seven models of the second set. We therefore used the functional form 'sex + SIMD quintile' consistently for all seven transitions of set two.

## Estimation of life expectancy and health-adjusted life expectancy

In a multistate survival model, total LE is defined as the sum of all state specific LEs, not considering years spent in the absorbing state [32]. State-specific LEs are independent of the start state and quantify the expected time spent in a state by a member of this population. This

accounts for the fact that not all members of the population start in the same state. In this study, we focused on total and state-specific LE at 50 years of age and assumed age 110 to be the maximum life span of individuals. We followed standard methodology to derive LE estimates from a multistate survival model (for example: [30, 32, 33]).

We first predicted start-state-specific probabilities of state occupancy, over age, for all unique combinations of covariates using each set of transition-specific models (see S3 Table for sets). From these probabilities we estimated start-state-specific LE, by summing up the respective probabilities over age.

We then weighted these start-state-specific LE estimates with the distribution of states at age 50 at the start of the study period ("weights"). These weights w~j can be directly interpreted as the prevalence of individuals with no, 1, 2 and 3+ complications at an average age of 50 on 01 January 2013 [32]. In order to obtain these weights for all relevant strata, we identified all individuals in the SCI diabetes database who were aged 45–54 years on 01 January 2013 to include in this distribution of start states to reduce random fluctuations due to low numbers. This meant that the average age of individuals used for the estimation of weights was age 50. While individuals aged 50–54 years were part of the actual study population, individuals aged 45–49 years were not. An overview of all utilized weights is presented in S4 Table.

LE at age 50 was then calculated as LE(50) = $\Sigma_j$ LE(50,j) * $w_j$(45–54), where LE(50,j) represents the start-state specific LE of state j and $w_j$(45–54) denotes the weight of state j.

We obtained 95% confidence intervals (95% CI) for all total and state-specific LE estimates using bootstrapping, by repeating the following steps 300 times. First, we drew a random sample from our study population, which was of the same size as our study population, using sampling with replacement. Second, we re-ran all analyses to estimate all total and state-specific LE. Third, we calculated standard errors for the 300 total and state-specific LEs which we used to calculate 95% CIs.

Data preparation and data analysis were carried out with *R* (Version 3.6.0) [34]. We used the R-package *flexsurv* [35] to estimate parametric Markov multistate survival models and to predict start-state-specific probabilities of state occupancy. A detailed example documenting our calculations is presented in S1 Text.

## Sensitivity analyses

In a sensitivity analysis we validated all LE estimates derived from our multistate models with two alternative approaches. We compared the multistate model estimates for LE with LE estimates obtained using a life table approach [36] and a parametric two-state Gompertz survival model [37]. This sensitivity analysis is the best-practice approach for validating the reliability of LE estimates obtained from multistate survival models [30, 38].

It is possible that differences in transition rates across SIMD quintiles were entirely driven by differences in the underlying distribution of health risk factors. In a further sensitivity analysis, we estimated a third set of models which included time-updated and mean-centered information on a number of health risk indicators, including HbA1c, total cholesterol, high-density lipoprotein-cholesterol, low-density lipoprotein-cholesterol, systolic blood-pressure, diastolic blood pressure, BMI, and information on smoking status. Parametric Gompertz survival models allow the covariate effect to be interpreted as a Hazard Ratio (HR). Therefore, we compared the HRs for sex and SIMD quintile from the second set of models with the HRs from this third set of sensitivity models.

## Results

### Overview of the study population

Table 1 provides an overview of the study population at study entry. We studied 4754 (55.3%) males and 3837 (44.7%) females with type 1 diabetes aged 50 and older on 01 January 2013. At study entry, individuals had a median age of 59.6 (*IQR*: 54.3 66.9) years and a median diabetes duration of 27.0 (*IQR*: 17.4 38.0) years. The median followed up time was 6.0 (*IQR*: 6.0 6.0) years.

### Number of complications

The most frequent complications were CKD stage 3 or worse (39.2%; 3370) and CVD (39.1%; 3363). Of those individuals with CKD, 257 individuals were in receipt of renal replacement therapy. At study entry, 38.1% (3274) of all individuals had none of the four complications. We note that of those defined as ever having had CKD by end of follow up, 83% had more than one eGFR < 60 mL/min/1.73 m^2 at least 3 months apart, thus meeting the KDIQO definition, and 39% had a history of moderately (3–30 mg/mmol) while 21% had a history of severely (>30 mg/mmol) increased albuminuria.

The proportion of individuals without complications decreased with age (Fig 2). This general trajectory was similar across all SIMD groups. In particular among the youngest studied

**Table 1. Characteristics of the study population of people with type 1 diabetes aged 50 and older in Scotland at study entry on 1st January 2013.**

| Summary | N / Median | Percentage / IQR |
|---|---|---|
| Males | 4754 | 55.3% |
| Females | 3837 | 44.7% |
| Age (years): Median | 59.6 | (54.3 66.9) |
| Diab. Duration (years): Median | 27.0 | (17.4 38.0) |
| Follow-Up Time (years): Median* | 6.0 | (6.0 6.0) |
| Cardiovascular Disease (CVD) | 3363 | 39.1% |
| Retinopathy/Maculopathy | 1403 | 16.3% |
| Chronic Kidney Disease (CKD) | 3370 | 39.2% |
| Diabetic Foot | 773 | 9.0% |
| - Number of Complications - | | |
| 0 Complications | 3274 | 38.1% |
| 1 Complication | 2657 | 30.9% |
| 2 Complications | 1870 | 21.8% |
| 3+ Complications | 790 | 9.2% |
| - SIMD 2016 Quintiles- | | |
| Quintile 1 | 1537 | 17.9% |
| Quintile 2 | 1838 | 21.4% |
| Quintile 3 | 1849 | 21.5% |
| Quintile 4 | 1619 | 18.9% |
| Quintile 5 | 1748 | 20.4% |

Note:

* Out of all 8,591 individuals, 6,769 individuals were followed for the entire 6-year period. The corresponding mean follow-up period was 5.4 (SD: 1.5) years. Out of all 1,822 individuals which were not observed for the entire 6-year period, 1495 individuals died during the follow up, and 327 individuals were censored as they became unobservable for other caused than death, such as out-migration.

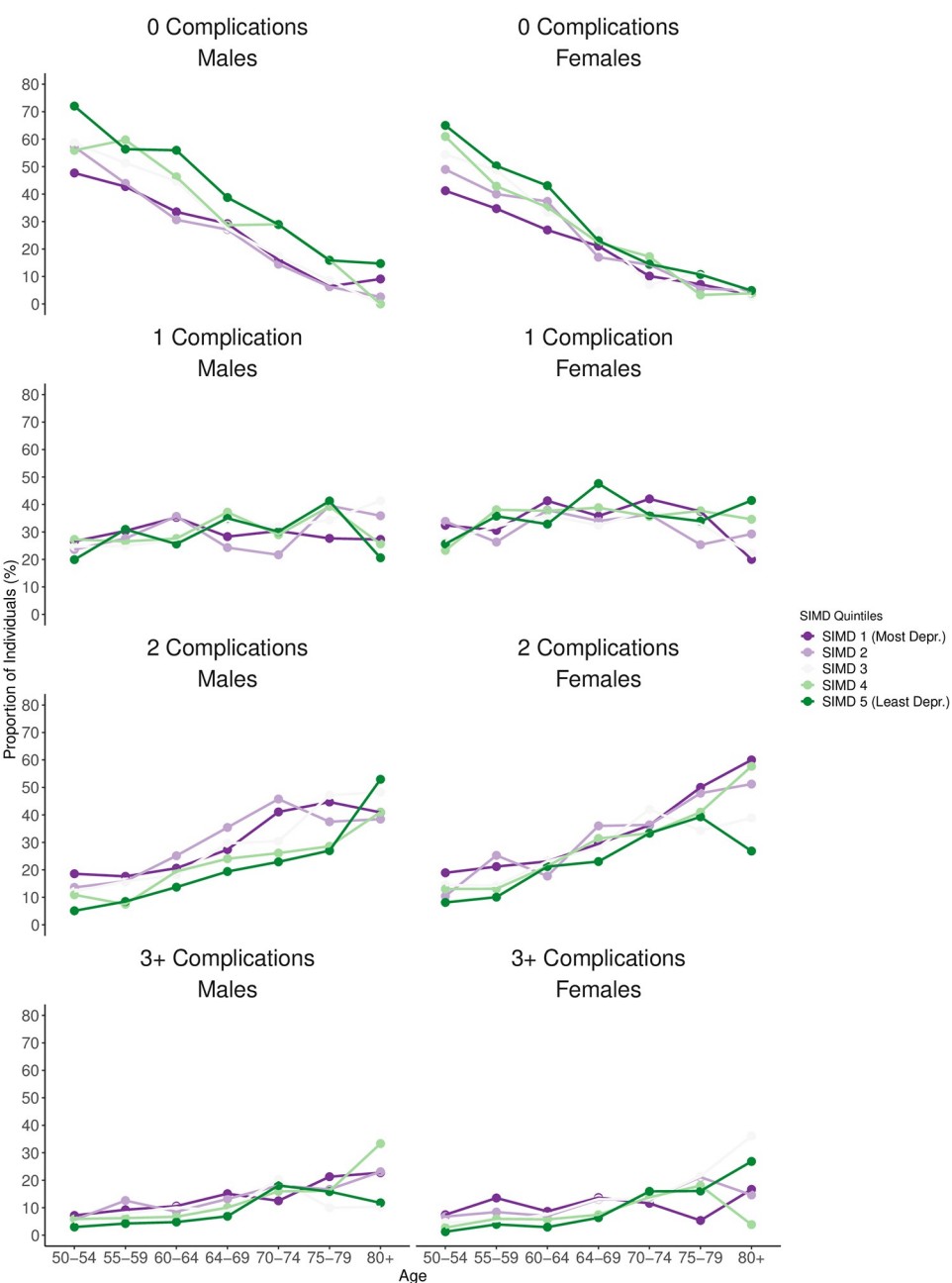

**Fig 2. Age-specific proportions of individuals with type 1 diabetes in the study population having no, 1, 2, and 3+ complications at point of study entry on 1st January 2013 by sex and SIMD quintile.**

age groups, we found a clear socioeconomic gap in the proportion of individuals without complications. For example, 72.0% of all males from the least deprived and 47.7% of all males from the most deprived quintile were free from complications at age 50–54. For females aged 50–54 from the least deprived quintile, 65.0% were free from complications compared to 41.2% from the most deprived quintile. For both, males and females from the most deprived quintile aged 50–54, the proportion without complications was similar to the proportion among the least deprived quintile who were at least 10 years older. Among males, the socioeconomic gradient

in the proportion of individuals without complications was relatively consistent over age. However, the corresponding gradient was not as equally consistent among females, in particular after the age of 65.

Among males and females, SIMD differences in the proportion of individuals with 1 complication were not always consistent across the studied age range. SIMD differences in the proportion of individuals with either 2 or 3+ complications over age were generally more consistent. We found that the proportion of males and females with either 2 or 3+ complications tended to be lowest among the least deprived SIMD quintiles while it tended to be highest among the most deprived SIMD quintile. However, SIMD differences with respect to 2 or 3+ complications were less strongly pronounced than for no complications and not always consistent at the oldest age groups.

A detailed overview of the prevalence of each studied complication for males and females of the study population at point of study entry is presented in S1 Fig.

### Transitions and mortality

Within the study period, we observed 4922 transitions. 3427 (69.6%) transitions occurred between the four transient states. A detailed summary of complications accounting for transitions between transient states is provided in Table 2. We found that cardiovascular disease was the single largest complication accounting for transitions between states of lower complication burden—in particular from no complications to 1 complication (Transition 1), and 1 complication to 2 complications (Transition 3). In contrast to this, diabetic foot was the single largest complication accounting for transitions among the highest complication burden: 2 complications to 3+ complications (Transition 5).

A summary of parameter estimates for all transition-specific models in provided in S5 Table (for set 1: 'sex') and S6 Table (for set 2: 'sex + SIMD quintile').

1495 (30.4%) transitions occurred into the absorbing state death. Fig 3 shows age-specific mortality rates for the study population within the study period. For males and females from all SIMD quintiles, the increase in mortality over age followed the same general trajectory. However, we found a clear socioeconomic gap in mortality at all ages. For example, for the age group 50–54, the mortality rate per 1,000 person years (PY) was 9.5 among males from the least deprived and 39.2 among males from the most deprived quintile. Among females in this age group, the mortality rate per 1,000 PY was 19.9 for the least deprived quintile compared to 29.9 for the most deprived quintile.

### Life expectancy and health-adjusted life expectancy at age 50

Total and state-specific LE estimates for all males and females are presented in Fig 4. LE at age 50 was higher among females (25.1 years (95% Confidence Interval: 24.4 − 25.9)) than among

**Table 2. Overview of complications accounting for transitions between transient states for the study population throughout the study period lasting from 01 January 2013 to 31 December 2018.**

| Transition | Retino./Maculo. | CVD | CKD | Diabetic Foot | All Transitions |
|---|---|---|---|---|---|
| No. 1: No → 1 | 342 | 405 | 303 | 96 | 1146 |
| Complication | (29.8%) | (35.3%) | (26.4%) | (8.4%) | (100%) |
| No. 3: 1 → 2 | 304 | 456 | 364 | 227 | 1351 |
| Complications | (22.5%) | (33.8%) | (26.9%) | (16.8%) | (100%) |
| No. 5: 2 → 3+ | 239 | 193 | 157 | 341 | 930 |
| Complications | (25.7%) | (20.8%) | (16.9%) | (36.7%) | (100%) |

*Note*: Transitions align directly to Fig 1—Panel (B).

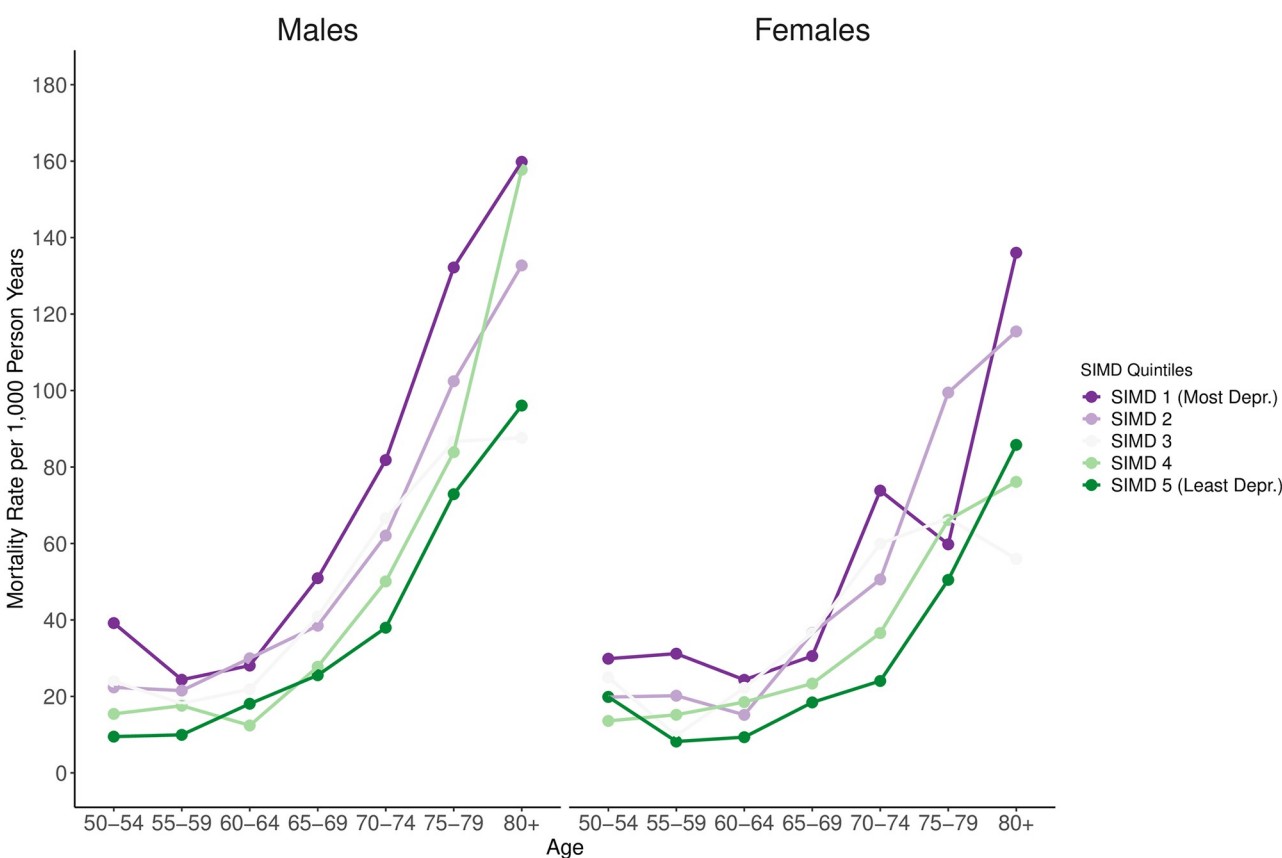

**Fig 3. Age-specific mortality rates for the study population during the study period by sex and SIMD quintile.**

males (22.2 years (95% CI: 21.6 − 22.8)). Females spent more years of life in all transient states than males, with the exception of life years spent with 3+ complications where years spent were similar.

Total and state-specific LE estimates for males and females by SIMD quintile are presented in Fig 5. LE was consistently highest among the least deprived quintiles and lowest among the most deprived quintiles. The gap in LE between the least and the most deprived quintile was 7.5 years among males and 8.1 years among females. At age 50, LE was 26.3 years (95% CI: 24.5 − 28.1) among males from the least deprived quintile, and 18.7 years (95% CI: 17.5 − 19.9) among males from the most deprived quintile. Corresponding levels among females were 29.3 years (95% CI: 27.5 − 31.1) for the least deprived quintile and 21.2 years (95% CI: 19.7 − 22.7) for the most deprived quintile.

Individuals from the most deprived quintile had the lowest LE and spent the fewest number of years without complications of diabetes. Differences in the number of years spent without complications, between the most and the least deprived quintile, were approximately 4 to 5 years among both males and females. At age 50, males from the least deprived quintile were expected to spend 9.3 years (95% CI: 7.5 − 11.1) without complications while males from the most deprived quintile were expected to spend 4.9 years (95% CI: 3.6 − 6.1) without complications.

Individuals from the least deprived quintile spent more years of life with 1 and 2 complications. For example, males from the least deprived quintile spent 8.0 years (95% CI: 7.0 − 9.0)

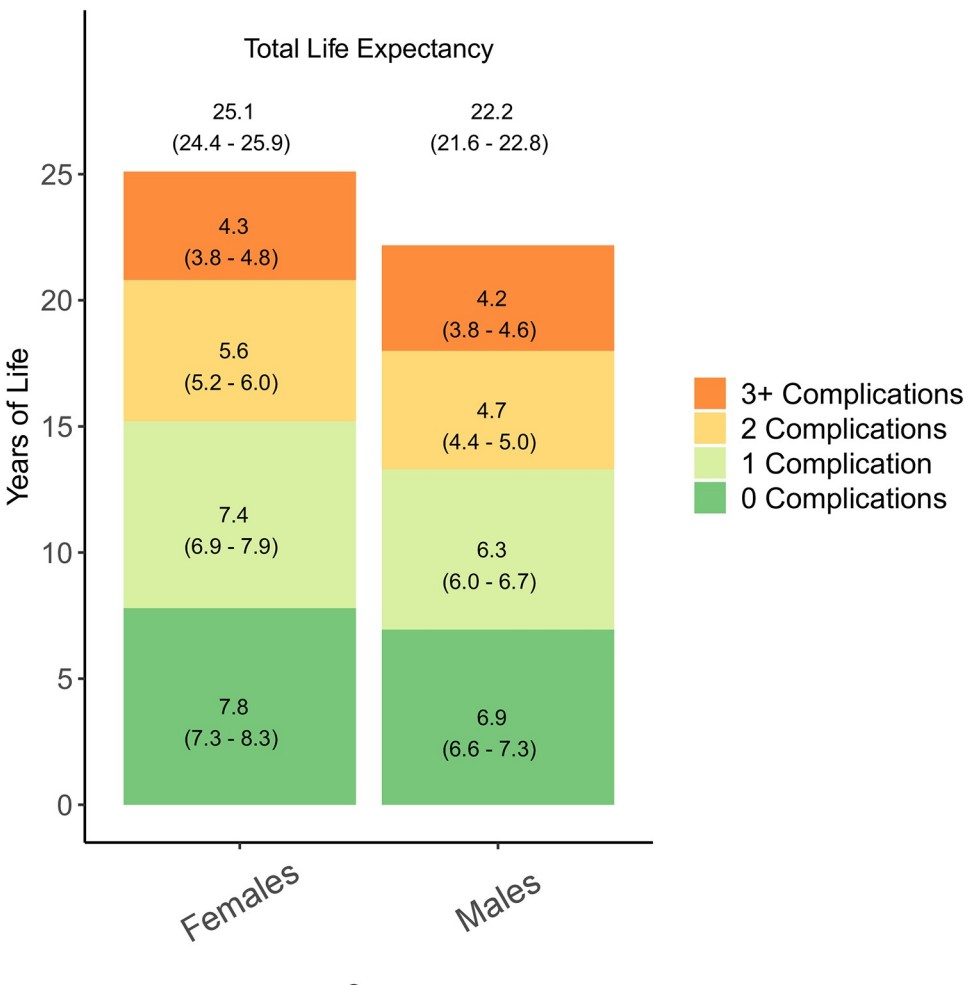

**Fig 4. Total and state-specific life expectancies at age 50 among males and females of the Scottish population with type 1 diabetes.**

with 1 complication, while males from the most deprived quintile spent 5.4 years (95% CI: 4.6 – 6.1) with 1 complication.

## Results of sensitivity analyses

Our life expectancy estimates were obtained from multistate models. All LE estimates obtained from multistate models were very close to results we obtained using two alternative approaches: the life table approach and a parametric two-state Gompertz survival model (S2 and S3 Figs). Total LE in a multistate survival model is calculated as the sum of all state-specific LE estimates—rather than being solely determined by transitions into the absorbing state death. The similarity of our multistate results to both alternative approaches provides a strong validation of our multistate model estimates. A small deviation in the results is common and was expected. We found that this deviation was slightly larger for males than for females.

We found differences in the distribution of risk factors across the studied SIMD quintiles as shown in the (S7 Table). For example, differences were especially prominent with respect to Hba1c levels and the fraction of ever smokers. Therefore, we examined whether differences in

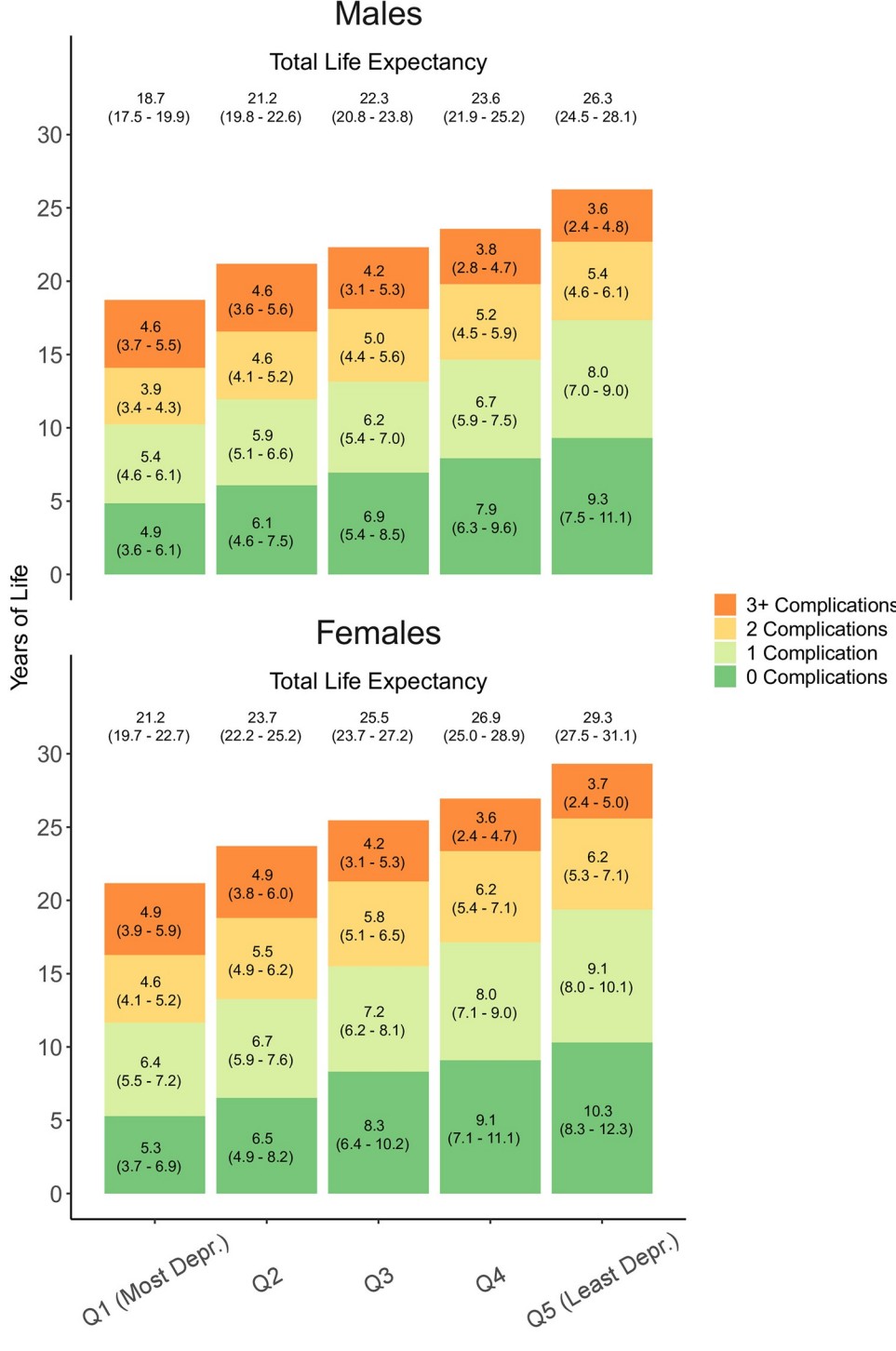

**Fig 5. Total and state-specific life expectancies at age 50 among males and females of the Scottish population with type 1 diabetes by SIMD quintile.**

transition rates for the different SIMD quintiles would persist when accounting for differences across health risk factors. We compared the covariate effects of 'sex' and 'SIMD quintile' from the second set of models with the corresponding effects from a third set of models, which included a number of health risk factors. Results of this sensitivity analysis are shown in S8 Table and illustrate that the HRs of the covariates 'sex' and 'SIMD quintile' changed only marginally when accounting for further health risk factors.

## Discussion

### Principal findings

Our results for the Scottish population with type 1 diabetes aged 50 years and older show a large socioeconomic gap in LE, as well as life years spent with and without complications of diabetes. We found that individuals with type 1 diabetes from the most deprived SIMD quintile experienced a double disadvantage: they had the lowest LE and spent the fewest number of years without complications of diabetes. The magnitude of socioeconomic differences in transition rates changed only marginally when controlling for differences in the distribution of health risk factors, demonstrating the wider role socioeconomic deprivation plays in determining health and mortality outcomes among the population with type 1 diabetes in Scotland.

### Comparison with the general population in Scotland

Clear parallels can be drawn when comparing our LE results by socioeconomic deprivation with LE results by socioeconomic deprivation reported for the corresponding general population in Scotland. Since the 1950s, LE among the general Scottish population has been among the lowest in Central and Western Europe [39]. LE for the population with type 1 diabetes in Scotland is lower than for the general population in Scotland. Previous results for LE at age 50 in the period 2008–2010 showed that LE was 7.4 years (95% CI: 6.5–8.3) lower for males and 9.0 years (95% CI: 8.0–10.0) lower for females with type 1 diabetes compared with the corresponding general Scottish population [16]. The multistate models we applied and the study period we covered means our LE results are not directly comparable to these previous findings for the Scottish population with type 1 diabetes or with routinely reported LE estimates for the general population. However, NRS LE estimates for the period 2014–2016, approximately the mid-point of our study period, provide a crude comparison. Comparing our LE estimates with these NRS estimates indicates that the gap in LE between the general population and the population with type 1 diabetes may have remained constant for males (7.6 years) but may have decreased for females (7.7 years) [40].

Socioeconomic inequalities in LE have been found for almost all countries and population subgroups in the world where appropriate data are available. Scotland is a country with a very steep mortality gradient between the most and least deprived [41]. Looking again at the closest comparable data for the general population, which used the same measure of deprivation and covered a similar time period, males aged 50 from the general Scottish population had a LE gap of 7.5 years between the most and least deprived quintiles [40]. For females aged 50 from the general Scottish population there was a LE gap of 6.1 years [40]. For males aged 50 with type 1 diabetes, we found a socioeconomic gap in LE of 7.5 years. This is very similar to the magnitude of the gap found for males of the general Scottish population. For females with type 1 diabetes, we found a socioeconomic gap in LE of 8.2 years, which may be larger than the gap for females of the general Scottish population [40].

To examine long-term LE trends among the population with type 1 diabetes and make direct comparisons with the general population, a different study design is required. A formal

trend analysis would need LE to be estimated consistently over time, using the same methodological approach.

The socioeconomic gap in healthy LE for the general Scottish population is estimated to be even higher than the socioeconomic gap in LE [42]. For the period 2015–2017, NRS reported a SIMD gap in healthy LE at age 50 of 13.2 years for males and 11.6 years for females of the general Scottish population [43]. Healthy LE is defined as the number of years lived in self-assessed good health and combines mortality with self-reported and subjective measures—typically from surveys [43]. This is not a comparable measure to our estimates of health-adjusted LE for the population with type 1 diabetes which combined mortality records and routinely collected information on the complications of diabetes. However, we found that the socioeconomic gap was largest for the years of life spent with no complications, the healthiest state possible for our study population.

## Comparison with previous findings for populations with type 1 diabetes

To our knowledge, no previous study has estimated health-adjusted LE for a real-world population with type 1 diabetes. We are also the first to estimate the socioeconomic gap in LE and health-adjusted LE for a national cohort with type 1 diabetes. Two previous studies provide a crude comparison of LE estimates for all males and females to the results we obtained in this study. In a previous study for the Scottish population with type 1 diabetes covering the period 2008–2010, LE at age 50 was 21.9 years for males and 23.2 years for females [16]. Despite minor differences in the methodology used, our results for the period 2013–2018 indicate that females have experienced larger gains in life expectancy than males (~ +1.9 years vs. ~ +0.3 years).

Levels of LE for males and females reported in our study were very similar to previous results reported for the population with type 1 diabetes in Australia at age 50 (Males: 22.7 years; Females: 25.9 years) [15]. The results for Australia are based on mortality data covering the period 1997–2010, which is around 10 years earlier than the period covered in our study. Despite the much earlier time period, levels of LE we find for males and females in Scotland are similar. This indicates that LE for males and females with type1 diabetes lags behind other countries and may be lower than LE is in Australia today. It is important to acknowledge that there are significant differences in life expectancy between the general populations of Australia and Scotland, evidenced for example in Human Mortality Database [44]. Therefore, it could be possible that other factors which are not directly associated with type 1 diabetes, such as Scotland's high levels of premature mortality and mortality from external causes of death [45], might explain parts of the LE gap between the Scotland and other cohorts of individuals with type 1 diabetes in other countries.

## Interpretations and implications

It is important to consider factors which are unique and specific in the context of type 1 diabetes. For individuals with type 1 diabetes, the risk of mortality is strongly associated with the number and duration of complications [46, 47]. Our results show that individuals from the most deprived quintile spend the fewest number of years without complications of diabetes. This is further evidence that deprived individuals experience complications earlier on in life and accumulate complications more rapidly.

Alongside overarching societal interventions targeting socioeconomic inequalities, keeping individuals with type 1 diabetes from deprived backgrounds free from complications for as long as possible is likely to have the biggest impact on the pathways driving the socioeconomic gap in LE. For example, preventing adverse short-term outcomes among individuals with type

1 diabetes, in particular events of hypoglycemia, DKA, and poor glycemic control, are key pathways for preventing adverse long-term health outcomes such as micro- and macrovascular complications or premature mortality [48, 49]. Therefore, interventions for narrowing the socioeconomic gap in long-term outcomes, including LE and health-adjusted LE, should consider targeting inequalities in short-term outcomes [50]. Patient-level interventions on the pathways to short term outcomes could include structured self-care programs and assertive outreach tailored specifically for more deprived socioeconomic groups and those disengaged with care, focusing on improved self-management, and minimizing disruptions to treatment regimen [51].

At the population level, interventions should reduce hurdles in accessing healthcare and ensure an equitable roll-out and uptake of technological innovations such as continuous sub-cutaneous insulin infusion (CSII) or flash monitors. Recent findings for Scotland suggested that the introduction of medical devices has been of greatest benefit for individuals with sub-optimal glycemic control [52], with individuals from the most deprived quintile being less likely to achieve glycemic targets [9]. Therefore, these technological innovations could contribute to reducing the socioeconomic gap in LE and health-adjusted LE for the population with type 1 diabetes. However, technological innovations may cause the socioeconomic gap to widen if uptake is unequally distributed.

## Strengths and limitations

We utilized routinely collected, electronic healthcare records, covering a national cohort with type 1 diabetes. Our results are therefore not affected by systematic recall bias and loss-to-follow-up. We operationalized four major complications of diabetes, but we did not capture additional health conditions such as cancers and cognitive impairments.

In the absence of unexpected health shocks such as wars and epidemics, LE tends to underestimate the average length of life for actual birth cohorts as it does not account for potential medical and technological progress in the future [53]. This means that the transition rates we observed are likely to change in the future. Therefore, our estimates of LE at age 50 may underestimate how long individuals aged 50 today will likely live for.

The way we captured the presence of all four complications reflects relatively advanced stages for each complication. Although a like-for-like comparison between each complication is not possible, we aimed to capture equally advanced stages. It is widely acknowledged that the results obtained from multistate models are sensitive to the way health and disease are measured [54]. It is therefore possible that other definitions would have led to either smaller or larger socioeconomic differences in life years spent with complications. We fully acknowledge that all four complications could have been defined differently. Considering this aspect, we acknowledge that our definition of CKD is less stringent than the KDOQI definition. Nevertheless, we argue that our definitions of complications provide reasonable estimates that do not over-estimate or under-estimate the magnitude of socioeconomic inequalities (see sensitivity analysis) or overemphasize one particular complication.

We defined the accumulation of complications as irreversible. In addition, we did not distinguish between multiple stages within each complication (i.e. stages of CKD) and we did not distinguish between grades of severity. As a result, our approach did not capture whether complications were temporarily reversed or progressed to more severe stages. While our multistate model is a clear simplification, it is in line with the highest-quality studies which have applied multistate survival models to obtain population-level metrics for broad population subgroups [30]. Studies specifically designed to examine the progression of one complication in more detail would require a specifically tailored focus and study design, including a much more

granular classification of each complication, the introduction of stages of disease progression, and an investigation of the predictive power of introduced covariates.

## Conclusion

Our findings for the Scottish population with type 1 diabetes aged 50 and older showed a large socioeconomic gap in LE and the number of years spent with and without complications. In addition to societal interventions, tailored support to reduce the impact of diabetes will be key for narrowing the socioeconomic gap in LE among the population with type 1 diabetes.

## Supporting information

**S1 Table. Definition of all diabetes-related complications examined in the study.**
(DOCX)

**S2 Table. Fraction of missing information for the study population (in percent) at study entry on 01 January 2013, before the usage of multiple imputation methods.**
(DOCX)

**S3 Table. Overview of all utilized transition-specific models used for the main analysis.**
(DOCX)

**S4 Table. Distribution of weights (in percent) at ages 45–54 by sex and SIMD quintile on 01 January 2013.**
(DOCX)

**S5 Table. Overview of all utilized transition-specific models of set 1.** Models of set 1 were used to derive estimates for all males and all females.
(DOCX)

**S6 Table. Overview of all utilized transition-specific models of set 2.**
(DOCX)

**S7 Table. Overview of the study population from SIMD quintile 1 (most deprived), 2, 3, 4, and 5 (least deprived) at point of study entry including all biomedical information utilized in the sensitivity analysis.**
(DOCX)

**S8 Table. Comparison of Hazard Ratios (HR) for sex and SIMD quintile from the utilized transition-specific models of set 2 with HR obtained from set 3.**
(DOCX)

**S1 Fig. Prevalence of the four studied complications retinopathy/maculopathy, cardiovascular disease, chronic kidney disease, and diabetic foot within the study population at point of study entry, separately for males and females and by SIDM quintile.**
(DOCX)

**S2 Fig. Comparison of LE estimates for all males and all females obtained from the multistate survival model (presented in the paper) with corresponding LE estimates derived using the Chiang (1984) Method and parametric two-state survival models.**
(DOCX)

**S3 Fig. Comparison of LE estimates for males and females by SIMD quintile obtained from the multistate survival model (presented in the paper) with corresponding LE**

**estimates derived using the Chiang (1984) Method and parametric two-state survival models.**
(DOCX)

**S1 Text. R-Code example demonstrating the estimation of health-adjusted LE.**
(DOCX)

**S1 Checklist. STROBE statement—Checklist of items that should be included in reports of *cohort studies*.**
(DOC)

## Acknowledgments

We would like to thank the Scottish Diabetes Research Network Epidemiology Group (SDRN-Epi): J. Chalmers (Diabetes Centre, Victoria Hospital, Kirkcaldy, U.K.), C. Fischbacher (Information Services Division, National Health Service National Services Scotland, Edinburgh, U.K.), B. Kennon (Queen Elizabeth University Hospital, Glasgow, U.K.), G. Leese (Ninewells, Hospital, Dundee, U.K.), R. Lindsay (British Heart Foundation Glasgow Cardiovascular Research Centre, University of Glasgow, Glasgow, U.K.), J. McKnight (Western General Hospital, National Health Service, Edinburgh, U.K.), J. Petrie and N. Sattar (Institute of Cardiovascular & Medical Sciences, University of Glasgow, Glasgow, U.K.), R. McCrimmon (Divison of Molecular and Clinical Medicine, University of Dundee, Dundee, U.K.), S. Philip (Grampian Diabetes Research Unit, Diabetes Centre, Aberdeen Royal Infirmary, Aberdeen, U.K.), D. McAllister (Institute of Health & Wellbeing, University of Glasgow, Glasgow, U.K.), E. Pearson (Population Health and Genomics, School of Medicine, University of Dundee, Dundee, U.K.), and S. Wild (Usher Institute, University of Edinburgh, Edinburgh, U.K.). The lead author of this group for this particular project is Helen M. Colhoun (helen.colhoun@ed.ac.uk).

## Author Contributions

**Conceptualization:** Andreas Höhn, Christian Dudel, Rosie J. Seaman, Helen M. Colhoun.

**Data curation:** Andreas Höhn, Stuart J. McGurnaghan, Thomas M. Caparrotta, Anita Jeyam, Joseph E. O'Reilly, Luke A. K. Blackbourn, Sara Hatam, Joseph Mellor, Naveed Sattar, Rory J. McCrimmon, Brian Kennon, John R. Petrie, Sarah Wild, Paul M. McKeigue, Helen M. Colhoun.

**Formal analysis:** Andreas Höhn, Rosie J. Seaman, Joseph Mellor.

**Funding acquisition:** Thomas M. Caparrotta, Helen M. Colhoun.

**Investigation:** Andreas Höhn.

**Methodology:** Andreas Höhn, Stuart J. McGurnaghan, Christian Dudel, Rosie J. Seaman, Helen M. Colhoun.

**Project administration:** Andreas Höhn, Naveed Sattar, Rory J. McCrimmon, Brian Kennon, John R. Petrie, Sarah Wild, Paul M. McKeigue, Helen M. Colhoun.

**Resources:** Stuart J. McGurnaghan, Luke A. K. Blackbourn, Sara Hatam, Naveed Sattar, Rory J. McCrimmon, Brian Kennon, John R. Petrie, Sarah Wild, Paul M. McKeigue, Helen M. Colhoun.

**Software:** Andreas Höhn, Stuart J. McGurnaghan, Luke A. K. Blackbourn, Sara Hatam, Paul M. McKeigue.

**Supervision:** Stuart J. McGurnaghan, Christian Dudel, Helen M. Colhoun.

**Validation:** Andreas Höhn, Stuart J. McGurnaghan, Thomas M. Caparrotta, Anita Jeyam, Joseph E. O'Reilly, Luke A. K. Blackbourn, Sara Hatam, Rosie J. Seaman, Joseph Mellor, Paul M. McKeigue, Helen M. Colhoun.

**Visualization:** Andreas Höhn.

**Writing – original draft:** Andreas Höhn, Thomas M. Caparrotta, Anita Jeyam, Joseph E. O'Reilly, Luke A. K. Blackbourn, Sara Hatam, Christian Dudel, Rosie J. Seaman, Joseph Mellor, Naveed Sattar, Rory J. McCrimmon, Brian Kennon, John R. Petrie, Sarah Wild, Paul M. McKeigue, Helen M. Colhoun.

**Writing – review & editing:** Andreas Höhn, Thomas M. Caparrotta, Anita Jeyam, Joseph E. O'Reilly, Luke A. K. Blackbourn, Sara Hatam, Christian Dudel, Rosie J. Seaman, Joseph Mellor, Naveed Sattar, Rory J. McCrimmon, Brian Kennon, John R. Petrie, Sarah Wild, Paul M. McKeigue, Helen M. Colhoun.

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
