## [Decision Letter · Decision Letter 0]

13 Oct 2021

PONE-D-21-20453Large socioeconomic gap in period life expectancy and life years spent with complications of diabetes in the Scottish population with type 1 diabetes, 2013-2018PLOS ONE

Dear Dr. Höhn,

Thank you for submitting your manuscript to PLOS ONE. After careful consideration, we feel that it has merit but does not fully meet PLOS ONE’s publication criteria as it currently stands. Therefore, we invite you to submit a revised version of the manuscript that addresses the points raised during the review process.

 The manuscript presents interesting data and is of potential interest given the scale of the analysis. However, several important limitations preclude to accept the manuscript in its current form. We invite you to submit a revised version of the manuscript that addresses the issues raised by the reviewer and the points indicated below.

1. Provide a more equilibrated view on the available literature in the "Introduction". You have stated "Estimates of LE [4–6] and health-adjusted LE [7,8] are available for the population with type 1 diabetes". However, after that the authors unexpectedly focused on the discussion of shortages of different studies presented by the references 9 and further. To provide equilibrated view on the literature, please discuss the findings described in the references 4-8 in this section of the "Introduction".

2. The phrase "As is typically observed in general populations, we hypothesized that LE would be higher among females than males". is unclear. What are the consequences of this hypothesis, and what for it has been introduced if you have real-world data that do not need any a priori hypotheses for the analysis?

In general, instead of a description of anticipated findings, please describe in the "Introduction" the major goals of your analysis and endpoints.

3. The authors described major steps in the analysis and supported them by the references to the previous publications in the "Data" section. However, this is not enough. Please provide enough details to make the self-sufficient description of the study population and data organisation. The authors could provide in the Supplement the exact list of  drug prescriptions and clinical information on the type of diabetes used for the selection of study sample. As of one of the major points of interest, please describe in details how you have identified the DM type (1 or 2) that is a difficult task the cohort of persons 50+ years old.

4. Regarding the cause of death analysis, please indicate whether you have used the multiple cause of death database available in Scotland, or classified a cause of death based on a single-reason approach.

5. Please indicate more details about "sociodemographic data and information on health status were obtained and continuously

115 updated using a long-format survival table capturing 30-day slices of time". Please indicate how you managed the analysis in the case of a person changed the residence from more prominent to less prominent social deprivation area defined by SIMD

6. The reference 20 intended to describe the calculation of SIMD does not contain enough bibliographic details to obtain the document and familiarise with the methodology of SIMD  calculation.

7. Please describe in details the ICD codes (or other codes or information classification) that have been used to obtain information about the four diabetes-related complications: retinopathy/maculopathy, cardiovascular disease (CVD), nephropathy, and neuropathy.

8. Regarding nephropathy, the author indicated only low eGFR and renal replacement therapy among the criteria. However, the first signs of diabetic nephropathy are presented by elevated albuminuria (or ACR ratio), while decreasing eGFR is rather late sign of diabetic nephropathy. Please describe whether the information about albuminuria is  available in the registry you used. Please also indicate which eGFR equation has been used, and most importantly - whether the confirmation of initial abnormally low eGFR has been considered to correctly classify a patient.

Please indicate how many patients had low GFR only, and how many recieved renal replacement therapy, instead of merging these highly heterogeneous conditions together.

9. Considering the importance of the state space and all seven distinct transitions for understanding the results of the analysis, please move the details provided in the  Figure S1 to the main manuscript. Please explain how "transient states can be entered and left" if previously it has been stated that the complications were considered irreversible, and confirmed this in a couple of sentences later "Transitions back to previous states were not allowed as we

164 assumed complications to be irreversible"? Please explain the meaning of the phrase "we did not allow transitions that skipped

165 one or more transient states".

10. At line 174, the authors indicated "We estimated two sets of seven parametric models.", but later described only two sets, with reference to the Supplementary containing description of 7 models. Please describe 7 models in brief also in the main manuscript.

11. I could not find the information on how you obtained the "weight of state j".

12. Please provide appropriate citations for the R packages used in the analysis, the PLoS ONE has no limits to the reference number.

13. The described approach is interesting and sophisticated. Please provide an example of the calculations performed for several patients included in the analysis (or several patients based on generated input data).

14. It is unclear why the authors did not include serum creatinine (and urinary albumin if available) among the list of biochemical and clinical parameters (HbA1c, total

219 cholesterol, high-density lipoprotein-cholesterol, low-density lipoprotein-cholesterol, etc) used for the modelling.

15. Please use one decimal sign for the vast majority of parameters during the presentation of descriptive statistics (individuals mean age of 61.40 has no too much meaning compared to 61.4). Please use Me and IQR for the description of diabetes duration and other variables with not normal distribution.

16. If only diabetic foot has been considered as a neurological complication, please use "diabetic foot " instead of "Neuropathy".

17. Please move the table(s) describing transitions in the main manuscript instead of pacing it in the Supplementary only.

18. Considering the most and least deprived quantiles, the authors have found almost 4-times difference in mortality rates expressed in person-years for males, but only 1.5-fold difference among females. However, the difference in the LE was not so prominent and was about 8 years for both males and females. Please explain the contrast results between PY-based mortality rates and LE metrics.

19. If only 40% have no complications at the initial point of the analysis, and the "LE was 26.26 years (24.47-

276 28.06) among males from the least deprived quintile, and 18.72 years (17.50-19.95) among

277 males from the most deprived quintile." it is rather unexpected to see that "males from the least deprived quintile were

286 expected to spent 9.31 years (7.50-11.12) without complications while males from the most

287 deprived quintile were expected to spent 4.85 years (3.64-6.07) without complications". Please check the correctness of calculations for the years spent without complications.

20. The discussion has to be substantially reviewed, including:

- appropriate placement of the "Limitations" section;

- appropriate consideration of literature indicated in the "Introduction" and other available world data;

- provision of hypotheses why diabetic complications in type 1 DM vary so prominently between social deprivation quintiles, while the Scotland has universal heath coverage that should reduce 4-fold difference in mortality rates. If the HRs of transitions changed only marginally when controlling for differences in the distribution of risk factors, which exactly mechanisms could be responsible for the observed differences between SIMD quintiles? I.e. if HbA1c concentration introduced in the model does not influence too much the outcome in persons with diabetes and thus supposed to be comparable between SIMD quintiles, which factors could be responsible for the differences? Moreover, this finding makes necessary to provide in a main text a table with all clinical and biochemical parameters analysed in this manuscript for each SIMD quintile;

- possible discussion of the literature considering the international studies aiming to reduce social inequalities for improving outcomes of DM type 2 and other (not DM) diseases.

21. The figures are nice, but more figures should be introduced to represent not only the frequency of the number of complications across ages and sex groups, but also the prevalence of each complication (cardiovascular, diabetic foot, kidney etc).

22. Please carefully revise all references, some of them have no year of publication (for example, ref 21), etc. Please submit your revised manuscript by Nov 27 2021 11:59PM. If you will need more time than this to complete your revisions, please reply to this message or contact the journal office at plosone@plos.org. Please include the following items when submitting your revised manuscript:A rebuttal letter that responds to each point raised by the academic editor and reviewer(s). You should upload this letter as a separate file labeled 'Response to Reviewers'.A marked-up copy of your manuscript that highlights changes made to the original version. You should upload this as a separate file labeled 'Revised Manuscript with Track Changes'.An unmarked version of your revised paper without tracked changes. You should upload this as a separate file labeled 'Manuscript'.

We look forward to receiving your revised manuscript.

Kind regards,

Boris Bikbov

Academic Editor

PLOS ONE

Journal Requirements:

All authors have completed and submitted the ICMJE Form for Disclosure of Potential Conflicts of Interest. TMC reports grants from Diabetes UK Grant: 18/0005786, during the conduct of the study. SHW reports meeting attendance supported by Novo Nordisk, outside the submitted work. RJM reports personal fees from Sanofi, personal fees from NovoNordisk, outside the submitted work. NS reports personal fees from Amgen, personal fees from AstraZeneca, grants and personal fees from Boehringer Ingelheim, personal fees from Eli Lilly, personal fees from Novo Nordisk, personal fees from Pfizer, personal fees from Sanofi, outside the submitted work; HMC reports grants, personal fees and other from Eli Lilly and Company, during the conduct of the study; grants from AstraZeneca LP, other from Novartis Pharmaceuticals, grants from Regeneron, grants from Pfizer Inc, other from Roche Pharmaceuticals, other from Sanofi Aventis, grants and personal fees from Novo Nordisk, outside the submitted work. No other disclosures were reported.

3. We noted in your submission details that a portion of your manuscript may have been presented or published elsewhere. [We confirm that this manuscript and its content is neither submitted/under review nor has been published elsewhere. An abstract with preliminary findings has been accepted for an oral presentation at the EASD 2021 Annual Meeting (28.09.2021 – 01.10.2021)] Please clarify whether this [conference proceeding or publication] was peer-reviewed and formally published. If this work was previously peer-reviewed and published, in the cover letter please provide the reason that this work does not constitute dual publication and should be included in the current manuscript.

5. One of the noted authors is a group or consortium [SDRN-Epi]. In addition to naming the author group, please list the individual authors and affiliations within this group in the acknowledgments section of your manuscript. Please also indicate clearly a lead author for this group along with a contact email address.

Reviewers' comments:

Reviewer's Responses to Questions

**Comments to the Author**

1. Is the manuscript technically sound, and do the data support the conclusions?

Reviewer #1: Yes

2. Has the statistical analysis been performed appropriately and rigorously? 

Reviewer #1: I Don't Know

3. Have the authors made all data underlying the findings in their manuscript fully available?

Reviewer #1: Yes

4. Is the manuscript presented in an intelligible fashion and written in standard English?

Reviewer #1: Yes

5. Review Comments to the Author

Reviewer #1: Hohn et al. conducted a retrospective cohort study that aimed to estimate life expectancy (LE) for the Scottish population with type 1 diabetes at age 50 and to examine how many years of life were spent with and without the most common complications of diabetes. Investigators used data from about 8000 individuals with T1D from the Scottish Care Information-Diabetes database, a nationwide register for diabetes. Main exposures associated with LE were the number of complications and the social gradient, assessed using the Scottish Index of Multiple Deprivation. As expected, investigators reported significant gaps in LE between individuals developing the most complications vs individuals developing few complications. Gaps in LE were also observed between most and least deprived areas.

This is an interesting study that provides further evidence on the impact of the social gradient on health, with a particular scope on individuals with T1D in Scotland.

Methods used in the papers appear to be state of the art approach. I particularly appreciated the quality of the transition-specific models.

However, a few elements need clarifications:

1. The SCID was implemented in 2004. However, January 1st 2013 was used as the study baseline. Investigators are invited to clarify why they did not include all individuals 50+y of age from 2004 instead of limiting the sample to 2013+. The sample size, with n=about 8000 appears to have been sufficient for the analyses, but it is likely that many more individuals could have been included. Please clarify.

2. How did authors manage potential change in SIMD for the same individual?

3. Authors suggest that they report a socioeconomic gap in the proportion of individuals without complications (Figure 1). While this is true for the 2 examples they mention in the text, mainly males/females aged 50-54, this difference is less evident as age increases. In females with 0 complication, from 64y+, there is no longer a marked socioeconomic gap. It if felt that the readers would benefit from this specification. Also, authors are invited to discuss the age X SIMD interaction in complication burden.

4. Figure 2 and elsewhere: authors are invited to present the number of individuals included in each of the 5 SIMD levels. Authors mentioned that the used SIMD quintiles: are they referring to quintiles within there study sample or quintiles within the SIMD index? While there are marked gaps between level 1 and level 5 exemplified in figure 2, it is unclear whether this reflect a significant proportion of the population or only extreme examples of poverty and wealth. I stress that I do not want to decrease the focus on the most deprived areas. However, such data would help to contextualize what are the socioeconomic gap authors are referring to.

5. Authors mention that the LE gaps they reported among individuals with T1D is similar to LE gaps in the general Scottish population. It would be interesting to develop more on this similarity. One would have expected that the LE gaps associated with T1D would have been more important than the general population.

6. PLOS authors have the option to publish the peer review history of their article (what does this mean?). If published, this will include your full peer review and any attached files.

Reviewer #1: No

---

## [Author Response · Author response to Decision Letter 0]

26 Jan 2022

Please find our responses to reviewers comments a separate file.

---

## [Decision Letter · Decision Letter 1]

22 Mar 2022

PONE-D-21-20453R1Large socioeconomic gap in period life expectancy and life years spent with complications of diabetes in the Scottish population with type 1 diabetes, 2013-2018PLOS ONE

Dear Dr. Höhn,

Thank you for submitting your manuscript to PLOS ONE. After careful consideration, we feel that it has merit but does not fully meet PLOS ONE’s publication criteria as it currently stands. Therefore, we invite you to submit a revised version of the manuscript that addresses the points raised during the review process.

Congratulations for the analysis and all work you have performed on the manuscript. The changes you have introduced and the provided explanations are very welcome and substantially improved the manuscript.

However, several issues are still remained and need to be addressed in the manuscript:

1. Regarding the rationale provided in the answers to the comment 8 (criteria for the CKD detection), there are following considerations:

- It remains unclear whether the 2 confirmations of initial abnormally low eGFR has been considered to correctly classify a patient, as required by the KDIGO guidelines.

- You referred to Krolewski 2015 to rationale that ACR lacks specificity and sensitivity for progressive decline in eGFR. Of note, the Krolewski's paper refers to DM type 1 while obviously the DM type 2 is predominant in the studied population. Moreover, the huge amount of other literature exists (including current guidelines) indicating that ACR is crucial in diabetic nephropathy evaluation. Please reflect these sources for providing equilibrated view.

- You referred to your own data indicating "The majority (59%) of those with chronic kidney disease stages G3–G5 did not have albuminuria on the day of recruitment or previously (Colombo et al. 2020)". Hopefully that means the ACEIs/ARBs work! (in constellation with genetic and other factors) GFR decline (especially 3a) could be related to age changes, and we also do not expect that a patient with diabetes should absolutely has diabetic nephropathy with elevated ACR. Moreover, non-diabetic nephropathies are prevalent in this population.

Finally, these considerations should be substantially revised. I didn't passed through the whole manuscript to see how you reflected this ideas, and in general it would be more easy to estimate the changes you have performed if you will cite them directly in Q&A sections, apart of highlighting the changed text over the 30 pages of the manuscript.

2. Please indicate how you managed the analysis in the case of a person changed the residence from more prominent to less prominent social deprivation area defined by SIMD. You explained that the 2016 release was used consistently

throughout the entire study period that is perfectly fine, but it remained unclear how was accounted a person who lived in place A with low SIMD and during the study transferred to place B with higher SIMD.

3. You provided very useful details for the states transition, but named the states just numerically. It would be much easier to provide some meaningful names instead of State 1 – State 3, etc.

4. Regarding the individual/population level considered in the comment 13 - I agree with your arguments, and completely understand that development of the individual-level decision making tool will require much more efforts and control. However, the population-level metrics are based on the individual-level estimates. You have proposed a very nice and interesting approach, but because of its complexity the exact calculations are not very clear. Due to this, please provide examples of 2-3 individual-level calculations - thus the readers could have more insights into the methodology.

5. Regarding data representation (Me and IQR vs X (SD)) - please leave normally distributed variables (like age) as X (SD). My comment for the Me and IQR concerns only the description of diabetes duration and other variables with not normal distribution. Regarding follow-up time over the entire 6-year period, please check the correctness of the calculations. The median 6.0 (IQR of 6.0 6-0) years would mean very low mortality, and inclusion of the majority of patients since from the beginning of the follow-up. Usually patients with diabetes are of high risk, and thus have elevated mortality, and more information of the reasons for the end of follow-up for 1822 (almost 20% of the studied population) should be provided, probably even better to present the mortality rate.

6. Regarding the life expectancy and mortality considered in the comment 18, it is very nice to see the literature-based example, but the raised issue was related to the exact findings in the manuscript. Moreover, in the answer you have referred to both "Remaining life expectancy was calculated at age 50" and "life expectancy at

birth" that somewhat confusing because only one indicator is considered in the manuscript. Finally, to resolve any doubdts, please provide in the manuscript the table with the calculations you made based on the literature data (i.e. mortality rates, life expectancy), but using instead of "Sweden" and "Kazakhstan" column data from the analysis for "males" and "females".

7. You provided very useful explanations in the comment 19, and there are no doubts about the robustness of the calculations. But please provide examples of 2-3 individual-level calculations - thus the readers could have more insights into the methodology of the multistate survival model. It seems that having the ready statistical code and all the data it will not take much time, but will demonstrate the readers how the exact numbers were calculated.

8. I could not find the figures titles in the main manuscript. Please check whether they are present in the file. Please also define clearly in the figures captions that LE refers to LE at 50 (both in the main manuiscript and the suppl).

9. You have produced very informative figures and tables for the Supplementary materials. However, each of them presented in a separate file that makes their downloading and revision extremely uncomfortable. Please prepare a single file with all Supplementary materials for the revision, you could use a free tool like pdfSam to merge all them.

Finally, I would like to congratulate you with the work you have performed for the sophisticated analysis of the very interesting data. Improving the several remaining points indicated above would be of great value for the future readers of the manuscript. Receiving from you the revised version, it would be possible to evaluate through the whole manuscript.

We look forward to receiving your revised manuscript.

Kind regards,

Boris Bikbov, MD, PhD

Academic Editor

PLOS ONE

Journal Requirements:

Reviewers' comments:

Reviewer's Responses to Questions

**Comments to the Author**

1. If the authors have adequately addressed your comments raised in a previous round of review and you feel that this manuscript is now acceptable for publication, you may indicate that here to bypass the “Comments to the Author” section, enter your conflict of interest statement in the “Confidential to Editor” section, and submit your "Accept" recommendation.

Reviewer #1: All comments have been addressed

Reviewer #2: All comments have been addressed

2. Is the manuscript technically sound, and do the data support the conclusions?

Reviewer #1: Yes

Reviewer #2: Yes

3. Has the statistical analysis been performed appropriately and rigorously? 

Reviewer #1: Yes

Reviewer #2: Yes

4. Have the authors made all data underlying the findings in their manuscript fully available?

Reviewer #1: Yes

Reviewer #2: Yes

5. Is the manuscript presented in an intelligible fashion and written in standard English?

Reviewer #1: Yes

Reviewer #2: Yes

6. Review Comments to the Author

Reviewer #1: I reviewed the revised version of the manuscript as well as the responses to the comments. All my questions have been adequately addressed. I have no further comments.

Reviewer #2: Thank you for the thorough response and changes. The revised manuscript is much stronger in the presentation of the methods and intepretation.

7. PLOS authors have the option to publish the peer review history of their article (what does this mean?). If published, this will include your full peer review and any attached files.

Reviewer #1: **Yes: **Jean-Philippe Drouin-Chartier

Reviewer #2: No

---

## [Author Response · Author response to Decision Letter 1]

4 May 2022

Please find our responses to the editors and the reviewers comments a separate file.

---

## [Editor Report · Decision Letter 2]

20 Jun 2022

PONE-D-21-20453R2Large socioeconomic gap in period life expectancy and life years spent with complications of diabetes in the Scottish population with type 1 diabetes, 2013-2018PLOS ONE

Dear Dr. Höhn,

Thank you for submitting your manuscript to PLOS ONE. After careful consideration, we feel that it has merit but does not fully meet PLOS ONE’s publication criteria as it currently stands. Therefore, we invite you to submit a revised version of the manuscript that addresses the points raised during the review process.

Congratulations for the very interesting analysis and all work you have performed on the manuscript. The changes you have introduced, the additional results are informative and made the manuscript better. Sorry for the delay in replying to your revised version. There are only several minor items that should be corrected because the PLOS ONE production team informed they require the very final version of the manuscript to proceed further. Please find below the list of minor items to be changes:

1. Please express numeric values of years of life and life expectancy with a complication with one decimal sign, since two decimal signs for years are excessive. For the same reason, I would suggest you to remove the "years" word from the parenthesis with the 95%CI (like "29.32 years (95% CI: 27.51-31.13 years)" and elsewhere - the second "years" seems excessive)

2. Line 62: please put the abbreviation KDIGO first, then the explanation.

3. In the Supplementary, please drop the "Comprehensive" word from the tables' titles.

4. In the STROBE statement I would suggest you to correct the referring from the pages number to the manuscript sections or limit the statement just with the "Yes" with possible details where necessary. Please note the final pdf version of the manuscript will have completely different page allocation, and the current version of the STROBE statement will not correspond to the pdf pages numeration. Another item concerning the STROBE Statement is the "Funding" details specified as "See online system", please be more specific.

5. Please change for the clarity the phrase introduced at lines 332-335, it is not clear why the threshold of 75 ml/min/1.73m2 has been selected in the part "with a further 10% having had a second record <75 /min/1.73m^2" - this value represents normal GFR in the absence of albuminuria. As a general idea apart of this manuscript, you have a nice data about the GFR and albuminuria that could be a topic of further manuscript based on the KDIGO-fully-compliant CKD definition, and also another analysis  investigating patients with a single abnormal GFR or albuminuria values.

Again, congratulations to the performed analysis.

We look forward to receiving your revised manuscript.

Kind regards,

Boris Bikbov, MD, PhD

Academic Editor

PLOS ONE
---

## [Author Response · Author response to Decision Letter 2]

23 Jun 2022

Please find our responses to all raised comments in a separate file attached to this submission.

---

## [Editor Report · Decision Letter 3]

24 Jun 2022

Large socioeconomic gap in period life expectancy and life years spent with complications of diabetes in the Scottish population with type 1 diabetes, 2013-2018

PONE-D-21-20453R3

Dear Dr. Höhn,

We’re pleased to inform you that your manuscript has been judged scientifically suitable for publication and will be formally accepted for publication once it meets all outstanding technical requirements.

Kind regards,

Boris Bikbov, MD, PhD

Academic Editor

PLOS ONE
---

## [Editor Report · Acceptance letter]

2 Aug 2022

PONE-D-21-20453R3 

Large socioeconomic gap in period life expectancy and life years spent with complications of diabetes in the Scottish population with type 1 diabetes, 2013-2018 

Dear Dr. Höhn:

I'm pleased to inform you that your manuscript has been deemed suitable for publication in PLOS ONE. Congratulations! Your manuscript is now with our production department. 

Kind regards, 

on behalf of

Dr. Boris Bikbov 

Academic Editor

PLOS ONE